# DRPBench: Evaluating LLMs in Concurrent Code Comprehension via Fine-Grained Data Race Prediction

**Yuqi Guo**[* 1 2]   **Siwei Wei**[* 1 2]   **Yan Cai**[1]

## Abstract

Large Language Models (LLMs) have demonstrated sophisticated comprehension of sequential code, yet their capacity for reasoning about concurrent programs remains largely unquantified. We introduce DRPBench, a benchmark designed to evaluate the concurrent code comprehension of LLMs by measuring their data race prediction performance. To address the challenge of runtime non-determinism for evaluation on concurrent programs, we frame the evaluation as a fine-grained static prediction task using 1,003 programs from the SV-COMP suite, featuring 557 manually annotated data races with precise variable- and line-level granularity. Our evaluation of 15 state-of-the-art LLMs—spanning standard, reasoning, and agentic variants—reveals that DRPBench effectively differentiates concurrent code comprehension capabilities of LLMs. While the top-performing model (Gemini 3 with test-time reasoning) achieves an F1 score of 74.89%, most models struggle significantly (scoring less than 60%), with Llama 3 70B achieving only 8.80%. Beyond benchmarking, we characterize two primary failure modes: (1) shared-variable distraction, where multiple variable appearances degrade comprehension accuracy, and (2) synchronization-logic myopia, the inability to interpret non-standard synchronization implementations. Our findings provide a diagnostic roadmap for enhancing concurrent code comprehension of LLMs in future development.

## 1. Introduction

Large Language Models (LLMs), such as the GPT family, Claude, and DeepSeek, have revolutionized software engineering. Their success is well-documented across diverse domains, including code completion (Ding et al., 2023; Liu et al., 2024; Yang et al., 2024b), bug prediction (Zhou et al., 2025; Yang et al., 2024a), and automated program repair (Bouzenia et al., 2025; Silva & Monperrus, 2025). As these models are increasingly integrated into the professional software development lifecycle, the demand for rigorous benchmarks to evaluate their code comprehension capabilities has grown. While the research community has responded with benchmarks ranging from basic function-level tasks (Bieber et al., 2020; 2022; Liu et al., 2023; Ding et al., 2024b) to complex repository-level challenges (Jimenez et al., 2024; Wang et al., 2025b), a critical dimension of modern software remains largely neglected: **concurrency**.

Concurrent programming is the backbone of modern computing, yet it remains notoriously difficult to master due to **runtime non-determinism**—where unpredictable scheduling causes a program's behavior to vary across executions (Lu et al., 2008). Robust comprehension of concurrent logic by LLMs could fundamentally alleviate the burden of writing and verifying concurrent software. Our study is motivated by this potential paradigm shift, providing the first rigorous assessment of whether LLMs can truly understand the non-deterministic concurrent execution.

The non-determinism of concurrent programs introduces a fundamental challenge for evaluation. Current benchmarks, such as CRUXEval (Gu et al., 2024) and RE2-Bench (Liu et al., 2025), rely on a fixed oracle (e.g., an execution output) to conduct the evaluation, where the model prediction is compared against the oracle to determine the correctness. However, this methodology is invalidated by concurrency, because a single input can yield multiple valid outputs across different executions due to the runtime non-determinism, making a unique oracle non-existent.

To address this challenge, we propose using data race prediction as a proxy for evaluating concurrent code comprehension. Identifying a data race requires an LLM to perform sophisticated code comprehension: it must recognize

---
[*]Equal contribution  [1]Key Laboratory of System Software (Chinese Academy of Sciences), Institute of Software, Chinese Academy of Sciences, China [2]University of Chinese Academy of Sciences, China. Correspondence to: Yan Cai <yancai@ios.ac.cn>.

*Proceedings of the 43rd International Conference on Machine Learning*, Seoul, South Korea. PMLR 306, 2026. Copyright 2026 by the author(s).

synchronization primitives, model the exclusive and causal relationships they enforce, and identify overlapping shared-memory accesses. Crucially, a data race is a static property of a program's logic; its existence is invariant to specific runtime scheduling. By framing concurrent code comprehension evaluation as a static prediction task, we provide a deterministic oracle for comparison and evaluation.

We introduce **DRPBench**, a benchmark designed to evaluate LLMs' code comprehension for concurrent programs. DRPBench is constructed from 1,003 concurrent programs sourced from SV-COMP (Jain et al., 2025), featuring 557 manually annotated data races with precise variable- and line-level granularity. DRPBench stands out among existing race prediction benchmarks by providing fine-grained evaluation. Unlike existing coarse-grained benchmarks (Jain et al., 2025; Liao et al., 2017; Chen et al., 2023b) that primarily focus on a program-level race prediction, DRPBench requires models to pinpoint specific variable names and line numbers for all the racing memory accesses, ensuring the evaluation reflects genuine logic comprehension rather than mere pattern matching.

Our evaluation of 15 state-of-the-art LLMs—spanning standard, reasoning, and agentic variants—reveals that concurrent code comprehension remains a significant challenge to existing models and can effectively differentiate LLMs based on their data race prediction capabilities. The top-performing model (Gemini 3 with test-time reasoning) achieves an F1 score of 74.89%, most models struggle significantly (scoring less than 60%), with Llama 3 70B achieving only 8.80%. Our in-depth analysis demonstrates that test-time reasoning (e.g., "thinking" mode) significantly improves performance by enabling multi-step causal reasoning. Conversely, integrating LLMs into agentic scaffolds (e.g., OpenHands (Wang et al., 2025a)) leads to performance degradation, suggesting that current agentic frameworks lack the specialized toolsets—such as thread-interleaving simulators—required to debug concurrent logic effectively.

Furthermore, we characterize two prevalent failure modes based on the common race prediction failures across different models: (1) **shared-variable distraction**, where high densities of shared variables degrade race prediction accuracy, and (2) **synchronization-logic myopia**, where the LLMs demonstrate an inability to generalize beyond standard library primitives to interpret custom synchronization implementations. We summarize actionable insights from the evaluation results and the failure modes for providing a diagnostic roadmap to the future development of code foundation models and agent scaffolds.

Our contributions are summarized as follows:

- **Deterministic Evaluation Proxy**: A methodology that leverages data race prediction to circumvent runtime

non-determinism, enabling reproducible assessment of LLMs' concurrent code comprehension.

- **DRPBench Dataset**: A new benchmark of 1,003 programs with 557 manually annotated variable- and line-level data races, providing the first fine-grained evaluation of LLMs' concurrent code comprehension.

- **Comprehensive Benchmarking**: An extensive study of 15 standard, reasoning, and agentic LLMs, demonstrating the strengths and limitations of current LLMs and agents in handling concurrent programs.

- **Diagnostic Roadmap**: Characterization of actionable insights and failure modes for future development of code foundation models and agent scaffolds.

**Conflict of Interest Disclosure:** None.

## 2. Related Work

**Benchmarks for Data Race Prediction.** The SV-Benchmarks (Jain et al., 2025), utilized in the annual SV-COMP competition, serve as a foundational resource for evaluating formal software verifiers. It includes a `NoDataRace` subtrack for evaluating software verifiers' capability to detect data races. Similarly, Dataracebench (Liao et al., 2017) is a benchmark that provides 208 OpenMP concurrent programs for race prediction evaluation and has been widely used in research studies (Chen et al., 2023a; Alsofyani & Wang, 2024; TehraniJamsaz et al., 2021; Lin & Liao, 2021; Lin et al., 2018; 2019; Shi et al., 2021). Other datasets have been synthesized by removing synchronization primitives from race-free programs or code snippets (TehraniJamsaz et al., 2021; Shen et al., 2025). However, these benchmarks primarily offer **program-level** evaluations, reducing the race prediction to a **binary classification** task. We argue that such coarse-grained evaluation methodology is insufficient to assess code comprehension; it may fail to distinguish between genuine semantic comprehension and superficial pattern matching. The Dataracebench-ML benchmark (Chen et al., 2023a;b) is the sole existing effort to incorporate line-level race evaluation. However, it still evaluates on a program-level pass/fail basis—requiring the line numbers of all races in a program to be correctly identified for a successful prediction—and is limited by its small scale (102 races) and low complexity (11–154 LOC, single shared variable in each program). **DRPBench** addresses these limitations by providing a dataset of high complexity and diversity with a fine-grained evaluation framework that evaluates individual variable- and line-level race localization. This helps us obtain in-depth insights into the LLMs' concurrent code comprehension.

**LLM Code Comprehension.** Evaluating how LLMs comprehend code execution is an active area of research. Exist-

ing efforts largely focus on code output prediction (Bieber et al., 2020; 2022; Liu et al., 2023; Ding et al., 2024b) or intermediate state reasoning (Ding et al., 2024a). While benchmarks such as HumanEval (Chen et al., 2021), MBPP (Austin et al., 2021), and CRUXEval (Gu et al., 2024) have pushed the boundaries of evaluating LLMs' sequential code comprehension, they fundamentally neglect the dimension of concurrency. To the best of our knowledge, DRPBench is the first benchmark to extend code comprehension evaluation to concurrent programs, requiring models to reason about thread interleavings rather than linear execution paths. It fulfills the research gap and provides a necessary perspective for evaluating the limits of current code foundation models and agents in concurrent code comprehension.

**Neural Approaches for Race Prediction.** Recent work has transitioned from heuristic and search-based methods to neural architectures for race prediction. Strategies range from early CNN-based classifiers (TehraniJamsaz et al., 2021) to advanced prompt engineering and parameter-efficient fine-tuning (Chen et al., 2023a; Shen et al., 2025). However, these studies remain coarse-grained program-level race prediction, i.e., predicting whether a program contains data races or not. This black-box prediction covers the specific failure modes of the models (e.g., whether a model missed a race due to synchronization complexity or variable density), hindering the future improvement of race prediction methods. The fine-grained evaluation methodology in DRPBench provides the diagnostic granularity necessary to move beyond simple classification toward explainable concurrent reasoning. We believe it can inspire future research studies in this field.

## 3. DRPBench

**Benchmark Overview.** DRPBench is constructed from 1,003 concurrent programs derived from the `NoDataRace` subtrack of the SV-Benchmarks (Jain et al., 2025). Originally designed for evaluating formal symbolic analyzers, the `NoDataRace` subtrack suite provides a high-quality corpus where each program is accompanied by a ground-truth binary label (racing vs. race-free). By annotating variable- and line-level race locations within the racing programs, our benchmark includes 226 racing programs containing 557 discrete data races and 777 race-free programs. The dataset features significant diversity in both origin and complexity:

- **Real-World Authenticity**: 621 programs are extracted from production-level projects, including the Linux kernel (drivers), Goblint, C-DAC, Deagle, DIVINE, and CProver.

- **Adversarial Hardening:** The remaining 382 programs are handcrafted by experts to feature subtle synchronization features, specifically designed to increase the difficulty of race prediction.

- **Diverse Scales:** On average, each program involves 219.80 thread instances and 4.66 shared variables. The program sizes vary from 14 to 624 lines of code, with a race density of 0 to 30 races per program.

Appendix A details these statistics and presents two example programs for demonstrating the diversity of race patterns.

**Fine-Grained Race Annotation.** To facilitate a rigorous assessment of whether LLMs possess a genuine semantic understanding of concurrent logic—as opposed to identifying races via superficial pattern matching—we performed extensive manual annotation of the 226 racing programs. For every individual data race, we annotated a metadata triplet $(\mathcal{V}, \mathcal{L}_1, \mathcal{L}_2)$:

1. **Variable Name($\mathcal{V}$):** The specific identifier of the shared variable being accessed concurrently without proper synchronization.

2. **Memory Access Locations ($\mathcal{L}_1, \mathcal{L}_2$):** The precise line numbers of the two conflicting memory accesses (e.g., Read-Write or Write-Write).

An illustration of this annotation format is provided in Figure 1. For the 777 race-free programs, the absence of races serves as the ground truth. We manually reviewed these race-free programs to ensure the correctness of the existing label. In total, the construction of DRPBench took over 500 person-hours of expert labor conducted over two months. This extensive human-in-the-loop validation ensures that the benchmark serves as a reliable ground truth for evaluating concurrent code comprehension in LLMs.

**Quality Assurance.** To ensure the high quality of race annotations, we employed a two-stage pipeline utilizing an ensemble of automated tools and human expertise. Specifically, we leveraged three distinct race prediction tools: OpenAI o3-mini (OpenAI, 2025) (a state-of-the-art reasoning LLM), GPT-4o (OpenAI, 2024) (a state-of-the-art non-reasoning LLM), and Deagle (He et al., 2022) (a symbolic analyzer and SV-COMP champion from 2022 to 2025). Figure 1 illustrates this annotation pipeline, which is structured as the following four steps:

1. **Independent Annotation.** Three human annotators and the three automated tools independently annotated data races within the dataset.

2. **Cross-Berification and Refinement.** Each human annotator was subsequently granted access to the tool-generated outputs. They were tasked with refining their initial annotations by specifically addressing discrepancies between their manual annotations and the tools' predictions.

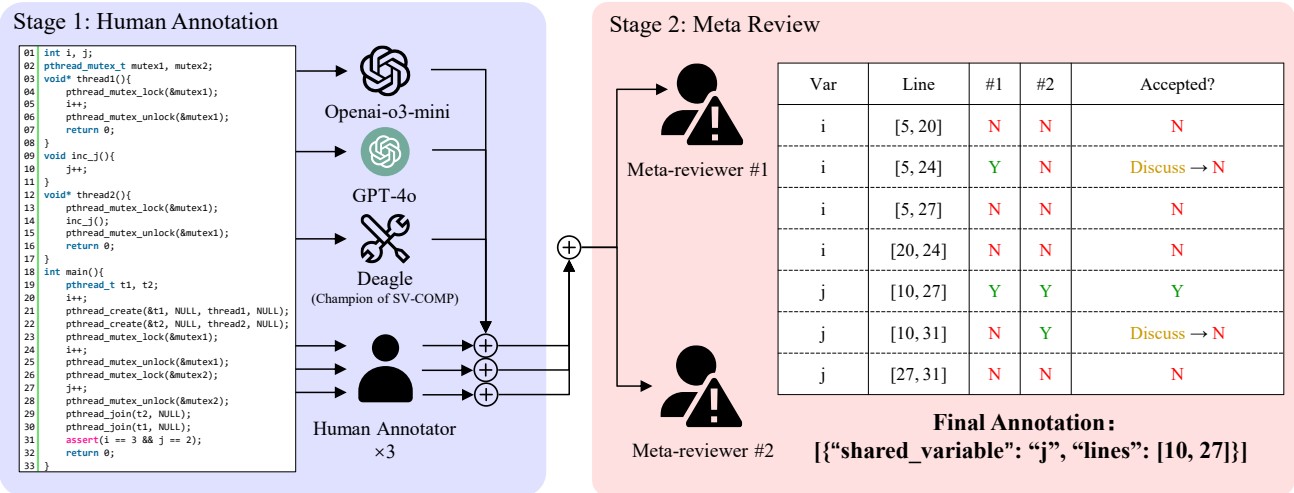

*Figure 1.* The pipeline of DRPBench's race annotation.

3. **Expert Meta-Review.** We introduced a rigorous meta-reviewing mechanism to synthesize the refined annotations. Two senior experts—each with over three years of professional experience in concurrent programming—independently evaluated the three sets of human-refined annotations and decided the correctness of every annotation.

4. **Consensus-Based Acceptance.** An annotation was accepted as the ground truth only if both meta-reviewers independently verified it as a true positive. Conversely, annotations denied by both experts were discarded. In cases of meta-reviewing disagreement, the experts engaged in a discussion to reach a consensus.

This collaborative process is designed to improve the quality of race annotation. As illustrated in Figure 1, out of the seven race annotations flagged by three human annotators and automated tools, four are directly rejected by unanimous meta-reviewing opinions, two are rejected after expert discussion, and only one sound data race constitutes the final annotation. This rigorous validation minimizes false positive annotations and ensures the benchmark serves as a reliable oracle for evaluating LLMs' concurrent code comprehension.

**DRPBench-Lite.** To facilitate a rigorous yet computationally efficient evaluation, we addressed two practical constraints: class imbalance and inference cost. The full DRPBench corpus is inherently skewed, with 777 race-free programs versus 226 racing programs. Furthermore, the high inference consumption of reasoning-variant and agentic LLMs makes full-scale evaluation prohibitive; for instance, Claude Sonnet 4.5 incurs costs of approximately $0.50 per program in reasoning mode and $1.50 in agentic mode. To mitigate these issues, we constructed a balanced representative subset, designated as **DRPBench-Lite**. We utilized stratified random sampling to select 100 unique racing programs and 100 unique race-free programs, ensuring a 1:1 class ratio. This balanced distribution prevents the evaluation metrics from being biased by the majority class. Unless otherwise specified, all subsequent empirical evaluations and error analyses are conducted on DRPBench-Lite.

**Prompt to LLMs.** We used a carefully crafted zero-shot prompt to instruct the LLMs to conduct the data race prediction. The prompt was structured into four sections, including:

1. **Task Definition.** This section defines the role and the task for the LLM, specifying that the goal is to predict data races within the given program.

2. **Domain Knowledge.** This section provides the definitions of data races and the concepts of synchronization primitives to ensure the LLM understands the domain. We evaluate the influence of this domain knowledge on model performance in Section 4.

3. **Output Specification.** This section instructs the model to present its race prediction, including the name of the racing variable and the line numbers of the conflicting memory accesses. We enforce a structured JSON schema to ensure deterministic parsing and facilitate automated evaluation. Specifically, the LLM is instructed to return a null or empty JSON object if the program is verified as race-free.

4. **Program Source Code.** The actual program to be analyzed, with a line number prepended at the head of each code line to facilitate unambiguous line number localization.

Due to space limitations, we present the prompt template in Appendix B.

**Evaluation Metrics.** To evaluate the granularity of the model's concurrent reasoning, we parsed the generated JSON objects and performed a set-based comparison against the ground-truth triplets. We defined that a predicted data race $\hat{r} = (\hat{v}, \hat{l}_a, \hat{l}_b)$ was classified as a **True Positive (TP)** if and only if there existed a ground-truth annotation $r^* = (v^*, l_1^*, l_2^*)$ such that:

$$(\hat{v} = v^*) \wedge (\hat{l}_a, \hat{l}_b = l_1^*, l_2^*)$$

This definition ensured that the model must identify both the correct variable name and the exact pair of conflicting access lines, ensuring unambiguous race localization. If a predicted race $\hat{r}$ did not satisfy the matching criteria for any $r^*$ in the program's ground-truth set, it was classified as a **False Positive (FP)**. If a ground-truth race $r^*$ was not matched by any predicted race $\hat{r}$ in the model's prediction set, it was classified as a **False Negative (FN)**. Based on these counts, we calculated three standard classification metrics: **Recall (R)**, **Precision (P)**, and the **F1-score (F1)**, defined as:

$$R = \frac{TP}{TP + FN}, \quad P = \frac{TP}{TP + FP}, \quad F1 = 2 \cdot \frac{R \cdot P}{R + P}$$

## 4. Evaluation and Analysis

We evaluated six state-of-the-art LLMs, encompassing both closed-source and open-source architectures: Gemini 3 (DeepMind, 2025), Claude Sonnet 4.5 (Anthropic, 2025), DeepSeek V3.2 (DeepSeek-AI, 2025), Qwen 3 32B (Qwen, 2025), Qwen 2.5 32B (Yang et al., 2025), and LLama 3 70B (Meta, 2024). To investigate the impact of architectural paradigms and inference strategies on concurrent code comprehension, we evaluated these models across three distinct configurations:

- **Standard**: We evaluated the basic race prediction capabilities by disabling the "thinking" mode or setting the reasoning effort to "low".

- **Reasoning**: We enabled test-time reasoning by enabling the "thinking" mode or setting the reasoning effort to "high".

- **Agentic**: For Gemini 3, Claude 4.5 Sonnet, and DeepSeek V3.2, we further evaluated their performance within the OpenHands (Wang et al., 2025a) scaffold. We did not evaluate Qwen 3 32B, Qwen 2.5 32B, and LLama 3 70B for this configuration because they did not support the tool calling function required by OpenHands.

All completion decoding parameters and agent configurations were maintained at their default settings to ensure a standardized baseline for comparison.

To provide a comprehensive performance contrast between neural architectures and traditional formal methods, we evaluated Deagle (He et al., 2022) and UGemCutter (Klumpp et al., 2022) on DRPBench-Lite. Deagle is a state-of-the-art static symbolic analyzer that has secured the championship in the SV-COMP competition for four consecutive years (2022–2025). It operates by encoding program execution paths and concurrency constraints into Satisfiability Modulo Theories (SMT) formulas, subsequently employing high-performance SMT solvers to identify potential data races. UGemCutter is a deterministic model checker for concurrent programs that has consistently placed in the top 3 of the SV-COMP ConcurrencySafety category (2022–2025). It combines the abstraction refinement-based verification algorithm trace abstraction with reasoning based on the principle of commutativity, i.e., the observation that some actions of different threads can be reordered without affecting program correctness. Based on commutativity, it reduces a given program (using partial order reduction algorithms) to a sound subset of interleavings and verifies the reduced program. Architecturally, these two formal methods are optimized for soundness and precision, prioritizing the minimization of false positives—a frequent challenge in automated program verification. By including these two artifacts, we established a rigorous performance ceiling for symbolic approaches, allowing us to quantify the relative comparison between LLMs and traditional symbolic approaches in analyzing concurrent programs.

Table 1 summarizes the performance of the evaluated models on DRPBench-Lite, ranked by their F1 scores. We append the suffixes -thinking and -agentic to denote configurations utilizing test-time reasoning and agentic scaffolding, respectively. For each metric, the top three results are highlighted. Our analysis of the empirical data reveals several key insights into the current state of LLMs' concurrent code comprehension:

**(1) Performance Heterogeneity and Benchmark Difficulty.** As summarized in Table 1, model performance across DRPBench-Lite varies significantly, confirming the benchmark's efficacy in distinguishing LLM capabilities. The top-performing model, Gemini 3-thinking, achieved the highest marks across all metrics ($R = 81.37\%$, $P = 69.36\%$, $F1 = 74.89\%$). However, even this frontier reasoning model leaves a considerable margin for improvement, particularly in precision. The performance gap between the top-tier and the remaining models is substantial. For instance, DeepSeek-V3.2-thinking and Claude-4.5-Sonnet achieved F1 scores of $58.24\%$ and $48.07\%$, respectively. Most notably, widely deployed models such as Qwen 2.5 32B and

*Table 1.* Evaluation results of state-of-the-art standard, reasoning, and agentic LLMs on DRPBench-Lite.

| MODEL | TP | FP | FN | | R | | P | | F1 |
|---|---|---|---|---|---|---|---|---|---|
| GEMINI-3-THINKING | 249 | 110 | 57 | #1 | 81.37% | #1 | 69.36% | #1 | 74.89% |
| GEMINI-3-AGENTIC | 227 | 207 | 79 | #3 | 74.19% | #4 | 52.27% | #2 | 61.33% |
| DEEPSEEK-V3.2-THINKING | 175 | 120 | 131 | #8 | 57.19% | #2 | 59.32% | #3 | 58.24% |
| GEMINI-3 | 197 | 186 | 109 | #6 | 64.38% | #5 | 51.44% | #4 | 57.18% |
| CLAUDE-SONNET-4.5 | 193 | 304 | 113 | #7 | 63.07% | #6 | 38.83% | #5 | 48.07% |
| QWEN3-32B-THINKING | 123 | 108 | 183 | #10 | 40.20% | #3 | 53.25% | #6 | 45.81% |
| CLAUDE-SONNET-4.5-THINKING | 248 | 811 | 58 | #2 | 81.05% | #10 | 23.42% | #7 | 36.34% |
| LLAMA3-70B-THINKING | 114 | 209 | 192 | #11 | 37.25% | #7 | 35.29% | #8 | 36.25% |
| DEEPSEEK-V3.2 | 135 | 343 | 171 | #9 | 44.12% | #9 | 28.24% | #9 | 34.44% |
| DEEPSEEK-V3.2-AGENTIC | 222 | 952 | 84 | #4 | 72.58% | #12 | 18.91% | #10 | 30.00% |
| CLAUDE-SONNET-4.5-AGENTIC | 221 | 1000 | 85 | #5 | 72.22% | #13 | 18.10% | #11 | 28.95% |
| QWEN2.5-32B-THINKING | 75 | 148 | 231 | #13 | 24.51% | #8 | 33.63% | #12 | 28.36% |
| QWEN2.5-32B | 62 | 253 | 244 | #14 | 20.26% | #11 | 19.68% | #13 | 19.97% |
| QWEN3-32B | 45 | 204 | 261 | #15 | 14.71% | #14 | 18.07% | #14 | 16.22% |
| LLAMA3-70B | 92 | 1694 | 214 | #12 | 30.07% | #15 | 5.15% | #15 | 8.80% |
| DEAGLE (SMT-BASED ANALYZER) | 113 | 42 | 193 | | 36.93% | | 72.90% | | 49.02% |
| UGEMCUTTER (MODEL CHECKER) | 44 | 4 | 262 | | 14.38% | | 91.67% | | 24.86% |

Llama 3 70B exhibited F1 scores below 40%, with standard Llama 3 70B struggling significantly at 8.80%. These results underscore that concurrent code comprehension is not an emergent property of scale alone; rather, it requires specialized architectures or inference strategies. The wide distribution of scores demonstrates that DRPBench possesses the necessary dynamic range to evaluate future iterations of code foundation models, from standard LLMs to reasoning or agentic variants.

**(2) LLMs significantly outperform symbolic analyzers in recall but exhibit lower precision.** The best-performing LLMs, including Gemini-3-thinking and DeepSeek-V3.2-thinking, achieved recall scores exceeding 55%, far surpassing those of Deagle and UGemCutter, whereas they struggled to maintain the same level of precision as the two symbolic analyzers. The relatively low recall of Deagle and UGemCutter stems from a lack of inherent semantic comprehension. These tools operate by mapping program structures to formal symbols and/or SMT formulas. This approach is often brittle when encountering semantically simple but hard-to-model code structures—such as loops—which can lead to timeouts or failed proof generation. We provide an in-depth analysis of these modeling challenges in Appendix D. Conversely, LLMs demonstrate a remarkable concurrency-awareness, allowing them to identify diverse data races that symbolic tools might miss. However, their relatively lower precision compared to symbolic methods suggests an inherent weakness: the LLMs can correctly identify overlapping memory accesses, but frequently fail to fully comprehend the synchronizations. This divergence between LLMs and symbolic tools also suggests a potential path for future improvement. We hypothesize that neuro-symbolic integration—combining the broad semantic search capabilities of LLMs with the rigorous symbolic inference of symbolic algorithms—could mitigate the weaknesses of both paradigms, facilitating more reliable and exhaustive concurrent code comprehension.

*Table 2.* The comparison between the reasoning LLMs and their standard versions.

| MODEL | DIFF (REASONING - STANDARD) | | |
|---|---|---|---|
| | R | P | F1 |
| GEMINI 3 | ↑16.99% | ↑17.92% | ↑17.70% |
| DEEPSEEK-V3.2 | ↑13.07% | ↑31.08% | ↑23.80% |
| CLAUDE SONNET 4.5 | ↑17.97% | ↓15.41% | ↓11.73% |
| QWEN3-32B | ↑48.37% | ↑20.76% | ↑31.85% |
| LLAMA3-70B | ↑7.19% | ↑30.14% | ↑27.45% |
| QWEN2.5-32B | ↑4.25% | ↑13.95% | ↑8.39% |

**(3) Reasoning models consistently outperform standard variants.** As shown in Table 2, test-time reasoning significantly improved the models' capacity to handle concurrent logic. On average, the reasoning models exhibit a performance gain of 17.97% in recall, 16.41% in precision, and 16.24% in F1-score over their standard counterparts. Our qualitative analysis of the generated solution chains (Appendix C) indicates that this performance shift is driven by a fundamental transition in problem-solving strategy. While standard models rely on surface-level heuristics—typically checking for the presence of a common mutex—reasoning models engage in active reflection and utilize self-correction to refine their understanding of the concurrency semantics.

We identified one notable exception to this trend: Claude 4.5 Sonnet. While its recall improved, its precision suffered a marked decline when enabling reasoning mode. Upon closer inspection, we found that the standard Claude 4.5 model suffered from severe spatial hallucinations: it frequently reported races on source lines that contained no memory accesses. In reasoning mode, this hallucination effect was exacerbated, leading to an increase in false positives. Despite this outlier, the overarching trend confirms that increased test-time reasoning is a primary driver for improving LLMs' concurrent code comprehension.

**(4) Current LLM agents are ill-equipped for concurrent program analysis.** As detailed in Table 3, while agentic

*Table 3.* The comparison between agentic LLMs and their standard versions.

| MODEL | DIFF (AGENTIC - STANDARD) | | |
|---|---|---|---|
| | R | P | F1 |
| GEMINI 3 | ↑9.81% | ↑0.83% | ↑4.15% |
| DEEPSEEK-V3.2 | ↑28.46% | ↓9.33% | ↓4.44% |
| CLAUDE SONNET 4.5 | ↑9.15% | ↓20.73% | ↓19.12% |

*Table 4.* The comparison between agentic LLMs and their reasoning versions.

| MODEL | DIFF (AGENTIC - REASONING) | | |
|---|---|---|---|
| | R | P | F1 |
| GEMINI 3 | ↓7.18% | ↓17.09% | ↓13.56% |
| DEEPSEEK-V3.2 | ↑15.39% | ↓40.41% | ↓28.24% |
| CLAUDE SONNET 4.5 | ↓8.83% | ↓5.32% | ↓7.39% |

*Table 5.* The impact of domain knowledge on race prediction.

| MODEL | DIFF (W/ DK - W/O DK) | | |
|---|---|---|---|
| | R | P | F1 |
| GEMINI-3 | -2.61% | 0.95% | -0.40% |
| DEEPSEEK-V3.2 | 0.33% | -0.09% | 0.04% |
| CLAUDE SONNET 4.5 | 6.53% | -3.26% | -0.19% |
| GEMINI-3-THINKING | 14.71% | -2.72% | 5.32% |
| DEEPSEEK-V3.2-THINKING | -20.91% | 17.54% | 3.80% |
| CLAUDE-SONNET-4.5-THINKING | 4.91% | 2.75% | 3.82% |

LLMs (using the OpenHands scaffold) improved recall by 15.81% over standard counterparts, they suffered a 9.74% decline in precision and a 6.47% decline in F1-score. Furthermore, when compared to their reasoning counterparts (Table 4), agents exhibited a staggering 20.94% drop in precision and a 16.39% drop in F1-score. Upon inspection of the agent behaviors, we attributed this performance degradation to a tooling-environment mismatch due to the following observations:

- In our experiments, the agents averagely performed 6.26 tool calls per program to conduct the race prediction, including 3.28 calls to the file editor, 1.62 calls to the thinking logger, 0.97 calls to the task summarizer, and 0.39 calls to the bash terminal.

- The agents primarily relied on in-context code analysis and static reasoning to predict data races. They frequently utilized the file editor as a persistent workspace to document candidate race-condition pairs and, while primarily using the thinking logger for recording internal reasoning steps during the trajectory. They occasionally employed terminal commands (e.g., grep) to search for specific shared variables and synchronization primitives in the source code. We present an example of agent trace in Appendix E.

- The agents primarily used the Python interpreter (accounted for in the bash terminal calls) as a postprocessing tool to ensure that the final race predictions strictly adhered to the required JSON schema. We do not observe that they employed it for data analysis or hypothesis grounding.

- Although specialized analysis tools like the gdb debugger were installed in the bash environment, LLMs lacked the operational mastery to utilize them for concurrent program analysis.

Consequently, the agents primarily default to simple file viewing and editing. Without constructive feedback from the execution environment, these multi-turn interactions served only to magnify spatial hallucinations, leading to the reported drop in precision. To bridge this gap, future research may focus on three strategic pillars: (i) developing concurrency-aware instrumentation and monitoring tools for agents, (ii) fine-tuning models on expert-level concurrent debugging traces, and (iii) optimizing agent workflows to maintain logical grounding over long-context analysis.

**(5) Reasoning LLMs effectively leverage domain knowledge to catalyze deductive accuracy, while standard models show negligible gains.** To evaluate the degree to which LLMs internalize the fundamental principles of concurrency, we conducted an ablation study comparing performance with and without a Domain Knowledge (DK) prompt section. This section provides formal definitions of data races and the concepts of common synchronization primitives. We focused this analysis on the three models: Gemini 3, DeepSeek V3.2, and Claude 4.5 Sonnet. As detailed in Table 5, our findings reveal a distinct divergence in how models utilize the domain knowledge. The inclusion of DK brought negligible influence on the standard models. This suggests that state-of-the-art LLMs already possess a saturated understanding of basic concurrency concepts. In contrast, reasoning LLMs leveraged the DK section to achieve an average 4.13% improvement in F1-score. While seemingly modest, this gain is significant given that it was achieved through simple prompt engineering without additional model finetuning. Manual inspection of the solution chains reveals that the DK section acts as a "reasoning anchor". Specifically, the causal and exclusive relationships suggested by the domain knowledge—including mutex semantics, semaphore signals, and thread lifecycle constraints—allowed reasoning models to ground their self-correction loops in causal logic rather than pattern recognition. This finding corroborates our previous observation that the reasoning capability is critical to concurrency code comprehension. For future development, we suggest industrial applications utilize carefully crafted prompt templates to activate the reasoning capabilities of code foundation models for better concurrent code comprehension.

Based on our evaluations, we propose the following actionable insights for researchers and developers:

**Actionable insights for LLM applications.** The evaluation of the race prediction task reveals the high potential of LLMs for comprehending concurrent code semantics. Notably, current frontier LLMs can outperform state-of-the-art symbolic program analyzers (like Deagle) in terms of recall, identifying a wider variety of true positive data races. However, this comes with severe hallucination. When applying LLMs to concurrent programs, the users must treat LLM outcomes as candidates rather than verdicts. Human-in-the-loop or symbolic verification is currently essential to filter false positives. A promising direction is the integration of LLMs with symbolic algorithms. This approach leverages the LLM's superior semantic comprehension to identify potential candidates while relying on the rigorous soundness guarantees of symbolic inference (e.g., SMT solvers) to eliminate hallucinations. Utilizing agentic tool-calling may also bridge the gap: agents can use feedback from internal consistency checks or external execution environments to refine their predictions.

**Actionable insights for LLM development.** The consistent superiority of reasoning LLMs over standard counterparts suggests that reasoning is a necessity for concurrent code comprehension. Future training efforts may focus on strengthening a model's ability to reason about concurrent thread interleavings. This likely requires datasets that include not just code, but execution traces and manual reasoning chains. For agents, a significant gap identified in our evaluation is the lack of an appropriate tool for handling concurrent programs. Future development may focus on developing tools that allow agents to control thread scheduling and monitor runtime shared memory accesses, or training LLMs specifically to use advanced debuggers (like gdb) to explore the thread interleavings of concurrent programs.

## 5. A Qualitative Analysis of the Failure Mode

To gain a deeper understanding of the specific weaknesses in current LLMs regarding concurrent code comprehension, we conducted a manual analysis of the race prediction failures that were common among different LLMs. These included the 56 false-negative and 22 false-positive cases where at least eight models failed to reach the correct prediction. From them, we characterized two failure modes.

**Shared Variable Distraction.** Our manual review of the 56 false-negative cases revealed a systemic weakness in how LLMs handle multiple occurrences of the same identifier. We found that the presence of multiple instances of a variable can distract the model's attention, leading to a failure in concurrent code comprehension. An example of this phenomenon is illustrated in Figure 2. In this instance, the program originally contains two accesses to the variable seed at Lines 13 and 17. A data race exists between the read at Line 13 and the write at Line 17. When introducing

```
01  inline int calculateNext(int s2)
02  {
03      int cnex;
04      do
05          cnex = rand();
06      while (cnex == s2 || cnex == 0);
07      return cnex;
08  }
09  int seed = 1;
10  inline int PseudoRandomUsingAtomic_nextInt()
11  {
12      int read, nexts;
13      assert(seed != 0);
14      atomic_acquire();
15      read = 1; // `read = seed;` leads to failure
16      nexts = calculateNext(read);
17      seed = nexts;
18      atomic_release();
19      return 0;
20  }
21  void *thr1(void *arg)
22  {
23      PseudoRandomUsingAtomic_nextInt();
24      return 0;
25  }
26  int main()
27  {
28      pthread_t t;
29      while (1)
30          pthread_create(&t, 0, thr1, 0);
31  }
```

*Figure 2.* Introduce the third instance of the seed variable, which can cause the LLMs to miss the data race between Lines 13 and 17.

the third access (a read) at Line 15, its presence causes the models to lose track of the unsynchronized relationship between Line 13 and Line 17. Analysis of the reasoning chains (Listings 5 and 6 in Appendix F) reveals that the models' attention appears limited to only two variable instances at a time. When introducing the third access at Line 15, the models can only focus on the last two appearances of the seed variable (Lines 15 and 17), which "mask" the earlier appearance at Line 13 and lead to a false negative.

**Synchronization-Logic Myopia.** Our analysis of the 22 false-positive cases revealed a systemic reliance of current LLMs on pattern recognition over causal semantic reasoning. Specifically, models struggle to recognize the synchronization when it is implemented via non-standard primitives rather than common library APIs. As illustrated in Figure 3, the program utilizes a custom exclusion protocol involving two shared variables, flag1 and flag2, alongside a standard mutex. This configuration ensures strictly exclusive access to the shared variable x between functions thr1 and thr2. The reasoning chain (Listing 7 in Appendix F) shows that the LLMs can correctly identify that the access to x is not protected by a pthread mutex. However, they fail to understand that the conditional loops regarding flag1 and flag2 implement an exclusive access, leading to a false positive report. When replacing the custom synchronization logic with standard pthread library calls, the LLMs can correctly identify the program as race-free. This suggests that current LLMs appear to utilize heuristic-based pattern recognition learned during pre-training to conduct concurrent code comprehension. When a program lacks the conven-

```
01  void *thr1(void *_) {
02      pthread_mutex_lock(&mutex);        // replace with the following
03      flag1 = 1;                         //   will be fine:
04      while (flag2 == 1)                 //
05      {                                  // pthread_mutex_lock(&flag1);
06          pthread_mutex_unlock(&mutex);  // pthread_mutex_lock(&flag2);
07          pthread_mutex_lock(&mutex);    //
08      }                                  //
09      pthread_mutex_unlock(&mutex);      //
10      x = 0;
11      return 0;
12  }
13  void *thr2(void *_) {
14      pthread_mutex_lock(&mutex);        // replace with the following
15      flag2 = 1;                         //   will be fine:
16      while (flag1 == 1)                 //
17      {                                  // pthread_mutex_lock(&flag2);
18          pthread_mutex_unlock(&mutex);  // pthread_mutex_lock(&flag1);
19          pthread_mutex_lock(&mutex);    //
20      }                                  //
21      pthread_mutex_unlock(&mutex);      //
22      x = 1;
23      return 0;
24  }
25  int main() {
26      pthread_t t1, t2;
27      pthread_create(&t1, 0, thr1, 0);
28      pthread_create(&t2, 0, thr2, 0);
29      pthread_join(t1, 0);
30      pthread_join(t2, 0);
31      return 0;
32  }
```

*Figure 3.* Non-standard synchronization implementations cause LLMs to generate false positive race predictions on the variable x.

tional "syntactic signatures" of synchronization—such as standard mutex or semaphore calls—the models default to a racing verdict, even when the underlying memory accesses are semantically synchronized.

## 6. Conclusion

We introduce a new benchmark, DRPBench, which is the first benchmark targeting the evaluation of LLMs' code comprehension on concurrent programs via the proxy of data race prediction. It consists of 1,003 concurrent programs and 557 fine-grained data race annotations. It decouples the LLMs' predictions to the granularity of individual data races for fine-grained evaluation. We conducted comprehensive experiments on 15 state-of-the-art standard, reasoning, and agentic LLMs, uncovering several key findings and insights. Additionally, we identified two common failure modes for LLMs' concurrent code comprehension. We believe these findings provide a diagnostic roadmap for future LLM research and development.

## Software and Data

The datasets, the evaluation scripts, and a README file for illustrating how to reproduce our evaluation results are available at https://github.com/guoyuqi020/DRPBench. We believe it can guarantee the reproducibility of our experiments.

## Acknowledgements

This research was supported by the Key Laboratory of System Software (Chinese Academy of Sciences), Institute of Software, Chinese Academy of Sciences. We thank Shihao Zhu, Xudong Zhang, and Luxia Lin for their time and effort in race annotation. We sincerely thank the reviewers for their invaluable feedback to improve our manuscript.

## Impact Statement

DRPBench is built upon the open-source benchmark SV-Benchmarks (Jain et al., 2025) and has undergone rigorous ethical reviews and content filtering processes to ensure compliance with the highest ethical standards. We take every precaution to guarantee that the code and data within DRPBench pose no risk of privacy leakage and meet all relevant legal requirements. This commitment ensures that DRPBench is not only an effective benchmark but also one that aligns with ethical guidelines, fostering both technical and ethical integrity.

We hope this work sets a new baseline for evaluating LLMs' capabilities in concurrent code comprehension and data race prediction. The dataset has the potential to inspire future research on leveraging NNs and LLMs for effective and efficient concurrent program analysis, including but not limited to data race prediction, thereby improving the software quality of concurrent programs. At this moment, we do not foresee any obvious undesirable ethical or social impacts.

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

*Table 6.* The statistics of DRPBench: the 226 programs that contain data races

| Category | N | Origin | Lines of Code | | | Number of Races | | |
|---|---|---|---|---|---|---|---|---|
| | | | Min | Max | Average | Min | Max | Average |
| goblint-regression | 56 | Goblint | 14 | 107 | 35.09 | 1 | 4 | 1.14 |
| ldv-races | 8 | Linux drivers | 65 | 150 | 103.13 | 2 | 7 | 3.5 |
| pthread | 20 | handcrafted | 42 | 140 | 69.45 | 1 | 4 | 2.00 |
| pthread-atomic | 10 | handcrafted | 47 | 322 | 120.90 | 2 | 30 | 12.30 |
| pthread-C-DAC | 1 | C-DAC | 61 | 61 | 61.00 | 2 | 2 | 2.00 |
| pthread-complex | 2 | handcrafted | 260 | 387 | 323.50 | 5 | 11 | 8.00 |
| pthread-deagle | 20 | Deagle | 28 | 55 | 39.25 | 1 | 2 | 1.75 |
| pthread-divine | 8 | DIVINE | 32 | 151 | 90.50 | 1 | 6 | 3.50 |
| pthread-driver-races | 4 | Linux drivers | 509 | 616 | 589.25 | 2 | 7 | 3.25 |
| pthread-ext | 44 | CProver | 33 | 246 | 103.34 | 1 | 30 | 2.32 |
| pthread-lit | 9 | handcrafted | 28 | 125 | 54.56 | 1 | 9 | 3.33 |
| pthread-nondet | 6 | handcrafted | 54 | 62 | 58.33 | 1 | 3 | 2.33 |
| pthread-race-challenges | 37 | Goblint | 25 | 79 | 45.84 | 1 | 8 | 1.41 |
| weaver | 1 | handcrafted | 88 | 88 | 88.00 | 2 | 2 | 2.00 |
| Summary | 226 | | 14 | 616 | 75.81 | 1 | 30 | 2.43 |

*Table 7.* The statistics of DRPBench: the 777 programs that are free of data races

| Category | N | Origin | Lines of Code | | |
|---|---|---|---|---|---|
| | | | Min | Max | Average |
| goblint-regression | 205 | Goblint | 15 | 170 | 66.70 |
| ldv-races | 19 | Linux drivers | 64 | 148 | 108.55 |
| pthread | 61 | handcrafted | 42 | 157 | 87.07 |
| pthread-atomic | 18 | handcrafted | 56 | 182 | 103.25 |
| pthread-C-DAC | 5 | C-DAC | 62 | 127 | 94.75 |
| pthread-complex | 6 | handcrafted | 156 | 391 | 275.75 |
| pthread-deagle | 24 | Deagle | 58 | 89 | 73.00 |
| pthread-divine | 10 | DIVINE | 32 | 39 | 35.5 |
| pthread-driver-races | 22 | Linux drivers | 509 | 624 | 601.00 |
| pthread-ext | 95 | CProver | 36 | 246 | 97.80 |
| pthread-lit | 14 | handcrafted | 40 | 85 | 55.40 |
| pthread-race-challenges | 63 | Goblint | 25 | 79 | 46.58 |
| pthread-wmm | 283 | handcrafted | 118 | 427 | 247.08 |
| weaver | 172 | handcrafted | 55 | 466 | 113.06 |
| Summary | 777 | | 15 | 624 | 159.49 |

# A. Statistics and Examples of DRPBench

Tables 6 and 7 present the statistics for the concurrent programs. The `category` indicates the name of the partition that the program belongs to within the SV-Benchmarks (Jain et al., 2025), `N` refers to the number of programs, and `origin` specifies the name of the project where the programs are sourced from. Table 8 presents the statistics for the data race patterns. To demonstrate the variety of programs and data races within DRPBench, we present two program examples in Figures 4 and 5. We illustrate their data races in the figure captions.

```
01  #include <pthread.h>
02  #include <assert.h>
03
04  pthread_mutex_t lock;
05  pthread_cond_t cond;
06  int x;
07  bool x_set = 0;
08
09  void *thread(void *arg)
10  {
11      (void)arg;
12      pthread_mutex_lock(&lock);
13      while (!x_set)
14          pthread_cond_wait(&cond, &lock);
15      assert(x == 42);
16      pthread_mutex_unlock(&lock);
17      return NULL;
18  }
19
20  int main()
21  {
22      pthread_t t;
23      pthread_create(&t, NULL, thread, NULL);
24      for (int i = 0; i <= 42; i++)
25          x = i;
26      x_set = 1;
27      pthread_cond_broadcast(&cond);
28      pthread_join(t, NULL);
29  }
```

*Figure 4.* The program contains a trivial data race on the variable x_set between lines 13 and 26. Interestingly, the variable x does not experience a data race, as the signal/wait mechanism and the while loop at line 13 ensure that the two accesses to x at lines 15 and 25 cannot run concurrently.

```
01  #include <stdlib.h>
02  #include <pthread.h>
03  #include <semaphore.h>
04  int data = 0;
05  sem_t data_sem;
06  void assume_abort_if_not(int cond)
07  {
08      if (!cond)
09      {
10          abort();
11      }
12  }
13
14  void *thread(void *arg)
15  {
16      sem_wait(&data_sem);
17      data = __VERIFIER_nondet_int();
18      sem_post(&data_sem);
19      return NULL;
20  }
21
22  int main()
23  {
24      sem_init(&data_sem, 0, 2);
25      int threads_total = __VERIFIER_nondet_int();
26      assume_abort_if_not(threads_total >= 0);
27      pthread_t *tids = malloc(threads_total * sizeof(pthread_t));
28      for (int i = 0; i < threads_total; i++)
29      {
30          pthread_create(&tids[i], NULL, &thread, NULL);
31      }
32      for (int i = 0; i < threads_total; i++)
33      {
34          pthread_join(tids[i], NULL);
35      }
36      free(tids);
37      return 0;
38  }
```

*Figure 5.* The program contains a data race on the variable data across multiple threads that concurrently reach line 17. Interestingly, the semaphore data_sem does not eliminate the data race, as it is initialized with a value of 2 (line 24), allowing at most two threads to enter the critical section (lines 16–18) simultaneously.

*Table 8.* Data race patterns.

| Category | Race Patterns | | Number of Threads | | | Number of Shared Variables | | |
|---|---|---|---|---|---|---|---|---|
| | Read-Write | Write-Write | Min | Max | Average | Min | Max | Average |
| goblint-regression | 22 | 42 | 2 | 30,000 | 3,221.06 | 1 | 300 | 4.31 |
| ldv-races | 10 | 18 | 2 | 2 | 2.00 | 1 | 2 | 1.68 |
| pthread-atomic | 99 | 24 | 2 | 4 | 2.89 | 2 | 7 | 4.06 |
| pthread-C-DAC | 1 | 1 | 2 | 8 | 3.20 | 1 | 5 | 2.80 |
| pthread-deagle | 35 | 0 | 2 | 51 | 19.29 | 1 | 8 | 2.58 |
| pthread-divine | 27 | 1 | 2 | 2 | 2.00 | 1 | 4 | 1.90 |
| pthread-ext | 88 | 14 | 2 | 321 | 9.11 | 1 | 6 | 3.53 |
| pthread-lit | 24 | 6 | 2 | 51 | 6.14 | 1 | 4 | 3.07 |
| pthread-nondet | 8 | 6 | 9 | 21 | 13.67 | 2 | 2 | 2.00 |
| pthread-race-challenges | 30 | 22 | 2 | 5 | 2.40 | 1 | 9 | 2.49 |
| pthread-wmm | 0 | 0 | 2 | 4 | 3.15 | 3 | 45 | 18.65 |
| pthread | 34 | 6 | 2 | 16 | 2.89 | 1 | 5 | 2.39 |
| weaver | 0 | 2 | 2 | 8 | 2.98 | 1 | 23 | 6.94 |
| pthread-complex | 9 | 7 | 2 | 7 | 4.17 | 1 | 11 | 6.83 |
| pthread-driver-races | 13 | 0 | 2 | 2 | 2.00 | 3 | 13 | 6.64 |
| Summary | 400 | 149 | 2 | 30,000 | 219.80 | 1 | 300 | 4.66 |

# B. Prompt Template

Listing 1 presents the prompt template used in our evaluation. The sentences colored in blue are for illustration purposes. They are not a part of the prompt template.

---

**Listing 1: Prompt template of DRPBench**

## Task definition
You are an expert at concurrent program design and data race detection. In the following, you will be given a program. You'll need to carefully look over the program to check whether it contains data race bugs. If it contains data race bugs, please locate them in line number pairs.
## Domain knowledge
The data race bug is a bug that occurs when (1) two or more threads access a shared variable at the same time, and (2) at least one of the accesses is a write. Note that, two operations **cannot** execute at the same time when (1) both are atomical operations, (2) both are protected by the same mutex, (3) they are guarded by a semaphare which ensures the exclusive access of the shared variable, or (4) other mechanism that forbids the two operations to execute at the same time.
The program can use `__VERIFIER_atomic_begin()` and `__VERIFIER_atomic_end()` to mark the start and the end of an atomic zone. Besides, if the function name has the `__VERIFIER_atomic` prefix, the corresponding function should also be regarded as an atomic zone. All operations inside the atomic zone should be regarded as atomic.
The program can use `pthread_mutex_lock(&m)` and `pthread_mutex_unlock(&m)` to lock and unlock a mutex `m`. The program can use `sem_wait()` and `sem_post()` to control semaphores; they do not lock or unlock mutexes. A semaphore holds an integer value. The `sem_wait()` is used to decrease the semaphore's value (typically by 1) to signal that the program wants to enter a critical section or use a resource. If the semaphore's value is greater than 0, `sem_wait()` decrements it and then proceeds. If the semaphore's value is 0, `sem_wait()` is blocked until the semaphore's value becomes greater than 0. The `sem_post` is used to increment the semaphore's value (typically by 1), indicating that a resource has been released.
The program can use `pthread_create()` to create a new thread and use `pthread_join()` to join the created thread. All the operations inside the new thread should happen after the `pthread_create()` site and before the `pthread_join()` site.
The program can use `pthread_cond_wait()` and `pthread_cond_signal()` to wait and signal a condition variable. It can also use `pthread_barrier_wait()` to wait for a barrier.
The program also uses `assume_abort_if_not()` as `assert()`. It can use `__VERIFIER_nondet_int()` to get a random integer. Besides, the indices of the lines are provided at the beginning of each line, e.g., "1:", to help locate the line numbers.
## Output specification
After thoroughly checking all potential data race bugs, please output all the confirmed data races. If no data race is found, please answer with an empty list. Please answer in the following JSON format (each race as one dict):
```json
{
    "races": [
        {
            "shared_variable": "the name of the shared variable",
            "lineA": the line number of the first memory access in `int` format,
            "lineB": the line number of the second memory access in `int` format
        },
    ...]
}```
## Source code
The code to be analyzed, with each line prepended by a unique line number.

---

## C. The Comparison Between Reasoning LLMs and Their Non-Reasoning Counterparts

This section presents a qualitative analysis comparing the solution chains of reasoning and standard counterparts to demonstrate that the observed performance superiority of reasoning models is primarily introduced by improvements in their underlying reasoning logic. We utilize the solution chains generated by DeepSeek-V3.2-thinking and DeepSeek-V3.2 from our main evaluation. We utilize the program illustrated in Figure 4 to substantiate this claim.

The solution chains for the case study program are presented in Listings 2 and 3. The solution chain of DeepSeek-V3.2 demonstrates a superficial detection strategy: it identifies data races simply by observing whether the shared variable is protected by a common mutex. This non-reasoning approach fails to integrate code semantics or execution logic, leading directly to the incorrect report of a false positive data race on variable $x$. In sharp contrast, the reasoning model, DeepSeek-V3.2-thinking, initially exhibits the same error, deeming the two accesses to variable $x$ to be a data race. However, it immediately engages in self-correction and reflection (signaled by the internal monologue, "But wait"). DeepSeek-V3.2-thinking then proceeds to detailly reason about the concrete concurrency semantics surrounding the accesses to $x$ and ultimately determines that the program semantics prevent the two accesses from running concurrently. This case study demonstrates that the observed performance superiority of reasoning models in race detection is primarily introduced by qualitative improvements in their underlying reasoning logic and the ability to self-refine.

---

**Listing 2: LLM completion from DeepSeek-V3.2-thinking**

(...)
Now, for x:
- The main thread writes x in line 25 (inside the loop) without any synchronization.
- The created thread reads x at line 15 (assert(x ==42)), which is inside the mutex.
So, the writes in line 25 are not protected, and the read in line 15 is protected. Since the main thread's writes are not synchronized, there's a data race between line 25 (write) and line 15 (read).
**But wait**, the main thread's loop writes x from 0 to 42. The created thread only reads x after x_set is 1. But when does x_set get set? The main thread sets x_set to 1 after the loop. So, the created thread's read of x (line 15) happens after x_set is 1.
(...)
But when does the created thread read x? Only after x_set is 1. And x_set is set to 1 after the loop. So, the writes to x in the loop are all before x_set is set to 1. Therefore, the created thread's read of x occurs after all the writes.
**Therefore, there's no data race on x**.

---

## D. The Limitation of Symbolic Tools

In DRPBench, the program can utilize a `while(true)` statement to start an "infinite" loop and then leverage a `if(...)` `break` statement to exit from the loop. It is extremely complex and usually impossible for symbolic tools to precisely infer how many iterations this "infinite" loop will execute. To proceed with the analysis, symbolic tools typically resort to heuristic estimations of the number of iterations, rules that are predefined by human experts. When these heuristics result in over-estimation or under-estimation, the analyzers will generate false positives or false negatives, respectively.

---

Listing 3: LLM completion from DeepSeek-V3.2

Let's analyze the program step by step to identify any potential data race bugs.

1. **Thread Creation and Joining**:
- The main thread creates a child thread at line 23.
- The child thread executes the thread function (lines 11-17).
- The main thread joins the child thread at line 28.

2. **Shared Variables**:
- x (line 6): an integer variable.
- x_set (line 7): a boolean flag indicating whether x has been set.
- lock (line 4): a mutex used to protect access to x and x_set.
- cond (line 5): a condition variable used to signal changes to x_set.

3. **Accesses to Shared Variables**:
- **Child Thread (thread function)**:
- Locks lock at line 12.
- Waits on cond (line 14) until x_set becomes true (line 13).
- Reads x at line 15 (protected by lock).
- Unlocks lock at line 16.
- **Main Thread**:
- Writes to x in a loop at line 25 (unprotected by lock).
- Writes to x_set at line 26 (unprotected by lock).
- Signals cond at line 27 (unprotected by lock).

4. **Potential Data Races**:
- The main thread writes to x at line 25 without holding the mutex lock. The child thread reads x at line 15 while holding the mutex. These accesses can happen concurrently, leading to a data race.
- The main thread writes to x_set at line 26 without holding the mutex lock. The child thread reads x_set at line 13 while holding the mutex. These accesses can also happen concurrently, leading to a data race.

5. **Synchronization**:
- The mutex lock is used correctly in the child thread to protect the accesses to x and x_set.
- However, the main thread does not use the mutex to protect its writes to x and x_set, which is incorrect.

6. **Conclusion**:
- There are two data races in the program:
- Between the write to x at line 25 and the read at line 15.
- Between the write to x_set at line 26 and the read at line 13.

# E. Agent Execution Trace

> **Listing 4:** An example of the agent trace
>
> **USER INPUT**
> The task prompt.
> **ACTION**
> The commands for reading the code file(s).
> **OBSERVATION**
> The content of the code file(s).
> **ACTION**
> AgentThink: Now I need to analyze this program for data race bugs. Let me carefully examine the shared variables and synchronization mechanisms. Let me analyze the program step by step: ...
> **OBSERVATION**
> The thought has been logged.
> **ACTION**
> AgentThink: Let me examine the functions more carefully to see which ones use synchronization: ...
> **OBSERVATION**
> The thought has been logged.
> **ACTION**
> AgentThink: Let me create a detailed analysis of all shared variable accesses and their protection: ...
> **OBSERVATION**
> The thought has been logged.
> **ACTION**
> AgentThink: Now let me identify the potential data races by checking which unprotected accesses can happen concurrently: ...
> **OBSERVATION**
> The thought has been logged.
> **ACTION**
> AgentThink: Let me also check if there are any races involving scull_quantum and scull_qset: ...
> **ACTION**
> The commands for writing the results, together with the content to be written
> **OBSERVATION**
> The file is created
> **ACTION**
> AgentThink: Let me verify the JSON file was created correctly:
> The commands for verifying the JSON structure.
> **OBSERVATION**
> The Python interpreter says that the JSON file is valid.
> **AGENT FINISH**

# F. Reasoning Chains for Failure Modes

---

**Listing 5: DeepSeek-V3.2-thinking's reasoning tragetory when the program contains only two *seed* instances.**

(...)

But the real shared variable here is seed. Let's look at where seed is accessed.

(...)

Wait, the code for PseudoRandomUsingAtomic_nextInt has:

Line 13: assert(seed !=0);

Line 14: atomic_acquire();

Line 15: read = 1;

Line 16: nexts = calculateNext(read);

Line 17: seed = nexts;

Line 18: atomic_release();

Wait, the code here seems to set read to 1, then compute nexts based on read (which is 1), then assign seed to nexts. **But seed is only being written here. The read from seed is line 14**, which is outside the atomic_acquire.

(...)

**So two threads could read seed in line 14 while another is writing it in line 17.**

(...)

---

**Listing 6: DeepSeek-V3.2-thinking's reasoning tragetory when the program contains three *seed* instances.**

(...)

Therefore, the accesses to seed in lines 15 (read) and 17 (write) are protected by this custom lock. So there's no data race here.

(...)

**But what about other accesses to seed? Are there any other lines where seed is accessed?** Let's check the code. The function calculateNext (line 1) checks if cnex is equal to s2 (which is the seed passed in), but that's a local variable. **The seed is read in line 15 and written in line 17. So all accesses to seed are within the critical section.**

(...)

---

**Listing 7: DeepSeek-V3.2-thinking's reasoning tragetory when the program contains non-standard synchronization implementations.**

(...)

Starting with flag1. Thr1 writes to flag1 at line 3 while holding the mutex. Thr2 reads flag1 at line 16. But wait, when thr2 is in the loop (line 16), it's inside the mutex lock. Because thr2 is holding the mutex when it checks flag1, and thr1 also locks the mutex when modifying flag1. So any access to flag1 (both read and write) is protected by the mutex. So no data race on flag1.

Similarly, flag2 is written by thr2 (line 15) under the mutex, and thr1 reads it at line 4. Again, both accesses are within the mutex, so no race there.

Now, the variable x. Thr1 writes to x at line 10, and thr2 writes to x at line 22. Are these writes protected by the mutex? Let's check.

In thr1, after exiting the loop, the thread unlocks the mutex (line 9) and then writes to x (line 10). Similarly, thr2 unlocks the mutex at line 21 and then writes to x at line 22. **So both writes to x happen after the mutex is unlocked. Therefore, the accesses to x are not protected by the mutex.**

---

# G. Limitation

**Data Diversity.** While the programs in the SV-Benchmarks cover a broad range of real-world applications and data race patterns, they cannot encompass every possible application and data race scenario. We aim to continuously expand the benchmark with new data races collected over time. Given that identifying and documenting data races is a time-consuming and labor-intensive process, our study primarily leverages existing datasets. Furthermore, DRPBench contains only isolated

pthread-based C/C++ concurrent programs, which may limit the generalizability of our findings to real-world software systems that often involve complex interactions between multiple components or are developed upon other programming languages (e.g., Java or Go). Our future work will extend the benchmark to include more recent data race vulnerabilities documented in sources such as the CVE[1] and CWE[2] databases.

**Memory Model.** Following the standard conventions of the SV-COMP (Jain et al., 2025), DRPBench currently assumes the **Sequential Consistency (SC)** memory model:

- The operations of each individual thread appear to the rest of the system in the exact order specified by its source code;

- The operations of all threads appear to be executed in some global total order, as if they were being processed by a single global switch that picks one instruction at a time from any thread.

Consequently, DRPBench does not currently include races that manifest only under a **Weak Memory Model (WMM)**, e.g., TSO or PSO. In future iterations, we intend to extend DRPBench to contain programs that manifest data races only under a WMM to increase the coverage of DRPBench.

**Limitations of LLMs in Race Prediction.** LLMs, by their nature, are "testers" rather than "provers"; they cannot formally prove the absence of a race. DRPBench is designed to evaluate this tester-prover boundary. By including non-trivial programs—where multiple conflicting memory access pairs exist but are correctly protected by complex synchronization logic—we measure the LLM's capacity to mitigate false positives. High precision on DRPBench thus serves as a critical proxy for an LLM's ability to recognize proper synchronization and correctly identify when a program is free of data races. However, it is important to note that this evaluation metric does not enable the evaluation of LLMs for their capability of proving that concurrent faults are absent. A potential future direction for addressing this limitation could involve integrating model-checking or data flow analysis algorithms into evaluation frameworks, such as Kripke structures, Trace Flow Graphs, or Petri nets.

# H. License

The DRPBench is licensed under the Creative Commons Attribution 4.0 International License (CC BY 4.0), whose content is summarized below.

# I. The Use of LLMs

This section clarifies the use of LLMs in our research study. In the dataset construction, we utilized the LLMs for helping to help annotate the data races (detailed in Section 3). In the evaluation experiments, we conducted experiments on open-source LLMs to assess their performance (detailed in Section 4). Additionally, we applied LLMs to polish the writing of this research paper.

---

[1]https://www.cve.org/
[2]https://cwe.mitre.org/

# Publication Agreement

This is a publication agreement[1] ("this agreement") regarding a written manuscript currently entitled

**DRPBench: Evaluating LLMs in Concurrent Code Comprehension via Fine-Grained Data Race Prediction**

("the article") to be published in PMLR ("the proceedings"). The parties to this Agreement are:

*Yuen Cui*

(name of corresponding author who signs on behalf of any other authors, collectively "you") and PMLR, ("the publisher").

1. By signing this form, you warrant that you are signing on behalf of all authors of the article, and that you have the authority to act as their agent for the purpose of entering into this agreement.

2. You hereby grant a Creative Commons copyright license in the article to the general public, in particular a Creative Commons Attribution 4.0 International License, which is incorporated herein by reference and is further specified at http://creativecommons.org/licenses/by/4.0/legalcode (human readable summary at http://creativecommons.org/licenses/by/4.0).

3. You agree to require that a citation to the original publication of the article in the proceedings as well as a hyperlink to the PMLR web site linking to the original paper be included in any attribution statement satisfying the attribution requirement of the Creative Commons license of paragraph 2.

4. You retain ownership of all rights under copyright in all versions of the article, and all rights not expressly granted in this agreement.

5. To the extent that any edits made by the publisher to make the article suitable for publication in the proceedings amount to copyrightable works of authorship, the publisher hereby assigns all right, title, and interest in such edits to you. The publisher agrees to verify with you any such edits that are substantive. You agree that the license of paragraph 2 covers such edits.

---

[1]The language of this publication agreement is based on Stuart Shieber's model open-access journal publication agreement, version 1.2, available at http://bit.ly/1m9UsNt.

6. You further warrant that:

   1. The article is original, has not been formally published in any other peerreviewed journal or in a book or edited collection, and is not under consideration for any such publication.
   2. You are the sole author(s) of the article, and that you have a complete and unencumbered right to make the grants you make.
   3. The article does not libel anyone, invade anyone's copyright or otherwise violate any statutory or common law right of anyone, and that you have made all reasonable efforts to ensure the accuracy of any factual information contained in the article. You agree to indemnify the publisher against any claim or action alleging facts which, if true, constitute a breach of any of the foregoing warranties or other provisions of this agreement, as well as against any related damages, losses, liabilities, and expenses incurred by the publisher.

7. This is the entire agreement between you and the publisher, and it may be modified only in writing. It will be governed by the laws of the Commonwealth of Massachusetts. It will bind and benefit our respective assigns and successors in interest, including your heirs. It will terminate if the publisher does not publish, in any medium, the article within one year of the date of your signature.

I HAVE READ AND AGREE FULLY WITH THE TERMS OF THIS AGREEMENT.

- Corresponding Author:
  - Signed:
  - Date:
    May 12, 2026