# OpenReview forum: "DRPBench: Evaluating LLMs in Concurrent Code Comprehension via Fine-Grained Data Race Prediction"
_ICML.cc/2026/Conference — ICML 2026 spotlight_

### Official Review · Reviewer_Y8fC · 2026-03-13

**Soundness:** 4
**Presentation:** 4
**Significance:** 4
**Originality:** 4
**Overall Recommendation:** 6
**Confidence:** 4

**Summary:**

The authors present DRPBench and extensively evaluate existing classic approach and LLM models using different agentic scaffolds. DRPBench consists of 777 race-free programs and 226 racing programs, and the lite version is sampled randomly and formed as 100:100 balanced dataset. The authors define the prediction as (v, l, l) and the evaluation metric as 1[(v, l, l)_pred = (v, l, l)] as the proxy to evaluate the capability of agentic system in finding data race condition.

**Compliance With Llm Reviewing Policy:**

Affirmed.

**Final Justification:**

The authors clarified my questions. I keep my current score.

**Key Questions For Authors:**

1. What happens when there are multiple race conditions in the same piece of code?
2. What are the tool use frequencies of agents? Do they use Python interpreter to write logics for grounding their hypothesis?

**Limitations:**

Yes, in Appendix G.

**Strengths And Weaknesses:**

Thanks for submitting to ICML 2026. I enjoy reading the paper and the paper itself indeed addresses an important problem in parallel software engineering.

### Soundness

Methodology: The authors clearly describe the curation of the dataset and the evaluation metric for benchmark. The representation of race condition tuple is sound and simple to evaluate.

Experiments: Extensive evaluation across different models (and their reasoning mode enabled/disabled) and agentic scaffolds are evaluated. Qualitative analysis is conducted to reveal agent's behaviors on race condition prediction.

### Presentation

I think the narrative is clear and easy to understand for general audience in SE/ML community. The tables and figures are neat.

### Significance

This benchmark addresses an important problem for LLM agents and defines the prediction task paradigm for such task. From the software engineering perspective, race condition prediction is a very critical problem in large complex system. This work contributes the baseline analysis for race condition prediction task with LLM approach and traditional approach, as the first step towards race condition prediction in sophisticated system.

### Originality

This is the first benchmark assessing if LLM understands race conditions. The paper provides sound problem statement for LLM, evaluation metric and the baseline results for both LLM (models and agentic frameworks) and traditional approaches.

---

> ### Author Rebuttal · Authors · 2026-03-30
>
> We are encouraged by Reviewer Y8fC’s recognition of DRPBench as addressing an important problem in parallel software engineering. We address the two key questions below.
>
>
> ### **Handling Multiple Race Conditions**
> For programs containing multiple race conditions (data races), we evaluate the model's performance using set-based retrieval metrics. Let $X$ be the set of ground-truth data races in a program, and $Y$ be the set of races predicted by the LLM. We calculate the metrics as follows:
> - $Recall = \frac{|X \cap Y|}{|X|}$
> - $Precision = \frac{|X \cap Y|}{|Y|}$
> - $F1 = \frac{2 \cdot \text{Precision} \cdot \text{Recall}}{\text{Precision} + \text{Recall}}$
>
> This granular evaluation ensures that an LLM is not over-scored for simply identifying that a program contains race conditions; it must precisely locate all the race condition pairs within the program while producing as few false positive predictions as possible.
>
>
> ### **Tool-Use Frequencies and Patterns**
> We analyzed the "Observation-Reasoning-Action" trajectories of the agents on DRPBench and observed the following:
> - On average, agents performed **6.26 tool calls per program**. This included 3.28 calls to the file editor, 1.62 to the thinking logger, 0.97 to the task summarizer, and 0.39 to the bash terminal.
> - Agents frequently utilized the file editor as a persistent workspace to take detailed notes on their intermediate analysis results, while primarily using the thinking logger for recording internal reasoning steps during the trajectory.
> - Instead of using a Python interpreter for hypothesis grounding, agents primarily relied on in-context code analysis and static reasoning. They used the file editor to document the candidate race condition pairs and occasionally employed terminal commands (e.g., grep) to search for specific shared variables and synchronization primitives within the source code.
> - While we observed calls to the Python interpreter (accounted for within the bash terminal calls), the agents primarily utilized the interpreter as a post-processing tool to ensure that the final race predictions strictly adhered to the required JSON schema.
>
>
> We thank the reviewer for these insightful questions and will incorporate the metric definitions and these detailed statistics into the revision of the manuscript.

---

> > ### Author Rebuttal · Reviewer_Y8fC · 2026-04-01
> >
> > Thanks the author for clarifications. I would like to keep the current score.

---

> > > ### Author Response · Authors · 2026-04-05
> > >
> > > We greatly appreciate your detailed and constructive review. Your insightful suggestions have been invaluable in enhancing the quality of our paper, and we will continue to refine the manuscript accordingly.

---

### Official Review · Reviewer_TsY7 · 2026-03-13

**Soundness:** 3
**Presentation:** 3
**Significance:** 2
**Originality:** 3
**Overall Recommendation:** 4
**Confidence:** 3

**Summary:**

This article proposes DRPBench to evaluate the understanding ability of large models for concurrent code. Specifically, due existing benchmarks provide simple and practical output results as a criterion for evaluating the correctness of understanding concurrent programs, it ignores the uncertainty that exists in the running of concurrent programs. To address this issue, DRPBench proposes using finer-grained data competition prediction as a criterion for evaluating LLMs’ ability to understand concurrent programs. DRPBench requires an LLM to accurately indicate the specific variable names where competition occurs and the exact line numbers of conflicting memory accesses.
DRPBench found that:

1. There is a huge difference in performance between models.
2. The reasoning mode (Thinking) plays a significant role.
3. The Agentic framework is currently performing poorly.
4. LLMs vs traditional symbol analyzers
Compared to traditional symbol analyzers, LLMsRecall has higher accuracy but lower precision.

Meanwhile, the author summarized the two major shortcomings of current LLMs.

1. Shared-variable distraction
2. Synchronization-logic myopia

**Compliance With Llm Reviewing Policy:**

Affirmed.

**Final Justification:**

The rebuttal addressed some of my concerns, but the concern about diversity remains. Therefore, my score didn't change.

**Key Questions For Authors:**

Please refer to the weaknesses section.

**Limitations:**

Yes

**Strengths And Weaknesses:**

**Strengths**
1. DRPBench is the first to evaluate the ability of large models to understand concurrent programs at a fine-grained level and propose a new evaluation benchmark.
2. The author conducted high-quality data annotation, combining automated tools with cross-validation by human experts in the annotation process to ensure the reliability of the evaluation benchmark.
3. The author evaluated mainstream models and systematically summarized the shortcomings of existing models: Shared-variable distraction，Synchronization-logic myopia.

**Weaknesses**
1. The types of programs included in the dataset are relatively limited, with only programs written in the C language, lacking diversity, which may limit the generalization ability of evaluation results.
2. There is no direct experimental comparison with existing evaluation benchmarks, so the effectiveness and advantages of DRPBench cannot be completely shown from the experiments.
3. The Agentic model performs worse, which seems abnormal. If some concurrent debugging tools are provided for these models, can they perform better?
4. Most of the author’s experiments only used DRPBench Lite. Since the entire dataset was proposed by the author, I believe it is necessary to conduct at least one complete experimental comparison.

---

> ### Author Rebuttal · Authors · 2026-03-30
>
> We thank Reviewer TsY7 for the thoughtful assessment of DRPBench’s rigor and high-quality annotation. We address the noted weaknesses below.
>
> ### **Program Type Diversity**
>
> We agree that expanding to other programming languages like Java or Go would enhance diversity. We chose C due to its prevalence in safety-critical systems and the availability of the verified SV-COMP suite. We will explicitly list "language diversity" as a focal point in our Limitations section.
>
>
> ### **Comparison with Existing Benchmarks**
> Unlike existing benchmarks  (e.g., SV-Benchmarks’ program-level "Race/No-Race" labels), DRPBench provides fine-grained line-level race location annotations. To demonstrate the necessity of this granularity, we evaluated the models on SV-Benchmarks and conducted the program-level race detection. In this setup, a prediction is marked as a True Positive (TP) if the LLM identifies at least one data race in a racing program, regardless of whether the reported location is correct. The evaluation results are presented below:
>
>
> Model | TP | FP | TN | FN | Recall | Precision | F1 |
> -------|----|----|----|----|--------|-----------|-----|
> Gemini-3-Thinking | 99 | 2 | 98 | 1 | 99.0% | 98.0% | 98.5% |
> Gemini-3-Agentic | 95 | 0 | 100 | 5 | 95.0% | 100.0% | 97.4% |
> DeepSeek-V3.2-Thinking | 91 | 11 | 89 | 9 | 91.0% | 89.2% | 90.1% |
> Gemini-3 | 97 | 31 | 69 | 3 | 97.0% | 75.8% | 85.1% |
> Claude-Sonnet-4.5| 98 | 44 | 56 | 2 | 98.0% | 69.0% | 81.0% |
> Qwen3-32B-Thinking| 77 | 17 | 83 | 23 | 77.0% | 81.9% | 79.4% |
> Claude-Sonnet-4.5-Thinking| 99 | 44 | 56 | 1 | 99.0% | 69.2% | 81.5% |
> Llama3-70B-Thinking | 82 | 23 | 77 | 18 | 82.0% | 78.1% | 80.0% |
> DeepSeek-V3.2| 83 | 51 | 49 | 17 | 83.0% | 61.9% | 70.9% |
> DeepSeek-V3.2-Agentic| 96 | 27 | 73 | 4 | 96.0% | 78.1% | 86.1% |
> Claude-Sonnet-4.5-Agentic| 100 | 28 | 72 | 0 | 100.0% | 78.1% | 87.7% |
> Qwen2.5-32B-Thinking | 70 | 23 | 77 | 30 | 70.0% | 75.3% | 72.5% |
> Qwen2.5 32B | 69 | 30 | 70 | 31 | 69.0% | 67.9% | 69.4% |
> Qwen3-32B| 68 | 17 | 83 | 32 | 68.0% | 80.0% | 73.5% |
> Llama3-70B| 88 | 84 | 16 | 12 | 88.0% | 51.2% | 64.7% |
> -------
>
> The evaluation results demonstrate that LLMs perform significantly better on SV-Benchmarks than on DRPBench. However, we contend that this high performance is **inflated**; our fine-grained evaluation in DRPBench reveals that models frequently fail to correctly locate race pairs despite identifying the program as "racing." This discrepancy indicates that LLMs often rely on probabilistic "sensing" or heuristic cues (such as the presence of shared variables) to predict races, rather than a rigorous understanding of concurrent interleavings. Consequently, DRPBench provides a higher-fidelity and more demanding assessment of true concurrent code comprehension than existing benchmarks.
>
>
> ### **Performance of Agentic Models**
>
> The reviewer identifies that agent performance is unexpectedly low. Our trajectory analysis (see our response to Reviewer Y8fC) shows that agents currently lack specialized concurrent debugging tools and rely almost entirely on static text reasoning. While providing concurrent debugging tools is a promising direction for improving agents' performance, to our knowledge, no current agent scaffold offers LLM-accessible concurrent debugging tools. DRPBench can serve as a vital benchmark to guide the future development of such tools.
>
> ### **Full Dataset Evaluation**
> To justify the use of DRPBench-Lite, we conduct a full-dataset evaluation on DRPBench using DeepSeek-V3.2-Thinking. We present the comparison of model performance on DRPBench-Lite and the full DRPBench as follows:
>
> - DRPBench-Lite: 57.19% Recall, 59.32% Precision, 58.24% F1.
>
> - Full DRPBench: 59.02% Recall, 60.58% Precision, 59.79% F1.
>
>
> The marginal difference confirms that DRPBench-Lite is a statistically sound and resource-efficient proxy for the full dataset, enabling researchers to achieve fast evaluation without sacrificing integrity.

---

> > ### Author Rebuttal · Reviewer_TsY7 · 2026-04-01
> >
> > Thanks to the author for providing additional information. Based on the results on SV Benchmarks and the explanation about the unexpected low performance of Agentic models, I am now confused as to why the Agentic models perform better on SV Benchmarks, while performing worse on DRPBench.

---

> > > ### Author Response · Authors · 2026-04-04
> > >
> > > We appreciate the opportunity to clarify this discrepancy. **The observation that agentic models appear superior on SV-Benchmarks but inferior on DRPBench is not an inconsistency in our data; rather, it highlights a critical evaluation gap that DRPBench is specifically designed to address.**
> > >
> > > **The discrepancy arises from the different evaluation metrics.**
> > > The SV-Benchmarks classifies the whole program as either a $TP_{prog}$, $FP_{prog}$, or $FN_{prog}$ based on whether the model correctly predicts the presence of data races.
> > > In contrast, DRPBench classifies every predicted race pair as either a $TP_{race}$, $FP_{race}$, or $FN_{race}$ based on the correctness of the race location.
> > > The correspondence between these two evaluation metrics is summarized as follows:
> > > - A $TP_{prog}$ corresponds to a collection of $TP_{race}$, $FP_{race}$, and $FN_{race}$.
> > > - A $FP_{prog}$ corresponds to one or more $FP_{race}$.
> > > - A $FN_{prog}$ corresponds to one or more $FN_{race}$.
> > >
> > > Given that all the three agentic models consistently outperform their non-agentic counterparts in terms of $TP_{prog}$ in SV-Benchmarks and $TP_{race}$ in DRPBench, we believe the reviewer's question refers to the counterintuitive phenomenon that the agentic models have a lower number of $FP_{prog}$ in SV-Benchmarks, whereas having a higher number of $FP_{race}$ in DRPBench.
> > > **We respectfully argue that this is not contradictory, but rather a natural consequence of the different evaluation metrics.**
> > > Due to the reason that the number of $FP_{race}$ can be affected by the number of both $TP_{prog}$ and $FP_{prog}$, it is possible for a model to simultaneously achieve a lower number of $FP_{prog}$ in SV-Benchmarks, while achieving a higher number of $FP_{race}$ in DRPBench.
> > > In our evaluation, the agentic models (especially Claude-Sonnet-4.5-Agentic and DeepSeek-V3.2-Agentic) report a significantly higher number of $FP_{race}$ on the $TP_{prog}$ programs than the non-agentic counterparts.
> > > This manifests as the observed performance discrepancy across the two benchmarks.
> > >
> > > In our original submission, we explained that this higher number of $FP_{race}$ was primarily caused by LLM hallucination.
> > > In our previous rebuttal post, we further analyzed that the agentic models tended to rely on static textual reasoning rather than a concurrent debugging tool, which explained why they suffered more severely than the non-agentic counterparts from hallucination.
> > > The SV-Benchmarks' coarse-grained evaluation reported a surface-level conclusion and thus masked the weakness of agentic models, leading to an incomplete evaluation of LLMs' concurrent code comprehension.
> > > **It was the fine-grained race-pair level evaluation of DRPBench that facilitated the finding of this deeply hidden weakness.**
> > > This makes DRPBench stand out from existing benchmarks.
> > >
> > > We hope this explanation resolves the confusion. Furthermore, we will add this clarification to the paper to help future readers better understand the advantages of DRPBench in evaluation metrics.

---

### Official Review · Reviewer_eN7h · 2026-03-13

**Soundness:** 3
**Presentation:** 4
**Significance:** 3
**Originality:** 3
**Overall Recommendation:** 5
**Confidence:** 5

**Summary:**

The paper introduces a benchmark, DRPBench, for evaluating LLMs ability for code comprehension of concurrent programs to detect race conditions. The paper is well-written, it contains a good introduction, literature review, description of the experiment design, experimental results and their interpretation. The results show a fairly low precision of LLMs in program analysis for concurrent faults. it is to be expected because to have high precision, ideally the proof of lack of a given concurrency fault such as a race condition or deadlock, the tool has to analyze possible interleavings. Model-checkers, especially those that perform non-deterministic execution such as Java Pathfinder shine in this area. An LLM bases its result on analysis of thread safety policies implementation (e.g. blocking to avoid a race condition), An LLM cannot evaluate interleavings by the nature of its inference method.

**Compliance With Llm Reviewing Policy:**

Affirmed.

**Final Justification:**

The rebuttal clarified my questions, in particular making a stronger connection to the performance of deterministic model checkers and pinpointing the limitations of deterministic program analysis tools. The authors have also clarified that DRPBench currently assumes the Sequential Consistency (SC) memory model.

**Key Questions For Authors:**

1. Have you considered using a non-deterministic execution based model-checker for ground truth generation similar to Java Pathfinder ? it might be more scalable than SMT based tools that involve solution of a set of inequalities that is very time consuming.
2. Some of the more subtle concurrency faults (including race conditions ) are due to memory model of the runtime system. For instance, in Java, the order of statements as observed from another threads is not guaranteed which may result in a race condition. This is described, for instance, in "Java Concurrency in Practice" by Brian Goetz. Does your benchmark include such non-obvious examples of race-conditions ?

E.g. the main method below may result in printing 0, 0 :

public static void main(String[] args) {
		// TODO Auto-generated method stub
		Thread one = new Thread(new Runnable(){
			public void run() {
				a=1;
				y=b;
			}
		}
		);
		Thread other = new Thread(new Runnable(){
			public void run() {
				b=1;
				x=a;
			}
		}
		);
		one.start(); other.start();
		try {
			one.join();
			other.join();
		}
		catch(Exception ex){ex.printStackTrace(); System.exit(1);}
		System.out.println("(" + x + "," + y + ")");
	}

**Limitations:**

As for limitations of the benchmark itself, I would suppose that it needs to incorporate some non-trivial race condition prone code fragments.
As for limitations of LLMs in detecting concurrency faults it is their inability to analyze iterleavings. Concurrency faults are due to different relative speeds of thread execution. An LLM provides its result just based on analysis of source code, just like testing, it will never be able to prove lack of concurrent faults. Model checking or data flow analysis can do that by analyzing interleavings in their models of computation (Kripke structures, Trace Flow Graphs, Petri nets etc)

**Strengths And Weaknesses:**

The main strength is a well designed experiment to evaluate LLMs ability to analyze implementations for concurrency faults.
The experiment design is sound. The presentation is well done. The significance is the results of experimentation on SV-COMP suite to evaluate LLMs ability to detect concurrency faults.

The DRPBench benchmark is original. It does not have direct counterparts.

---

> ### Author Rebuttal · Authors · 2026-03-30
>
> We thank Reviewer eN7h for the insightful feedback regarding the inherent limitations of LLM inference compared to formal model checking. We address the specific questions below.
>
>
> ### **Model Checking for Ground Truth**
>
> We agree that non-deterministic execution-based model checkers are highly effective for exploring interleavings. While we did not use a model checker for the initial race location annotations, we recognize the opportunity to use a formal model checker to validate and refine DRPBench’s annotations in future versions.
>
> To address the reviewer’s interest in model checkers, we evaluated UGemCutter on DRPBench-Lite. UGemCutter is a software model checker for concurrent programs that has consistently placed in the top 3 of the SV-COMP ConcurrencySafety category (2022–2025). On DRPBench-Lite (306 ground-truth races), UGemCutter detected 44 races and produced 4 false positives. Manual inspection confirmed these false positives were due to algorithmic limitations rather than configuration errors. Consequently, UGemCutter achieved 14.38% Recall, 91.67% Precision, and 24.86% F1. These results indicate that even state-of-the-art model checkers find the fine-grained localization in DRPBench highly challenging compared to SMT-based tools like Deagle.
>
> We sincerely appreciate Reviewer eN7h for raising this point, and we will integrate this evaluation into the final version of our manuscript.
>
> ### **Weak Memory Models**
> Following the standard conventions of the SV-COMP NoDataRace track, DRPBench currently assumes the **Sequential Consistency (SC)** memory model:
>
> - The operations of each individual thread appear to the rest of the system in the exact order specified by its source code;
>
> - The operations of all threads appear to be executed in some global total order, as if they were being processed by a single global switch that picks one instruction at a time from any thread.
>
> Consequently, DRPBench does not currently include races that manifest *only* under a **Weak Memory Model** (**WMM**, e.g., TSO or PSO suggested by the reviewer's example).
>
> We acknowledge this as a limitation. We will add a discussion in the revised manuscript regarding the challenges LLMs face with WMMs. This is a particularly high-impact direction for future iterations of DRPBench.
>
>
> ### **LLMs vs. Formal Proofs**
> We fully agree with the reviewer that LLMs, by their nature, are "testers" rather than "provers"; they cannot formally prove the absence of a race.
>
> DRPBench is designed to evaluate this tester-prover boundary. By including non-trivial programs—where multiple conflicting memory access pairs exist but are correctly protected by complex synchronization logic—we measure the LLM’s capacity to mitigate false positives. High Precision on DRPBench thus serves as a critical proxy for an LLM's ability to recognize proper synchronization and correctly identify when a program is free of data races. To provide a more rigorous context for the difference between LLM-based and traditional formal analysis, we will incorporate the reviewer's points on Kripke structures, Trace-Flow Graphs, and Petri nets into our Limitations section, contrasting these formal modeling techniques with the probabilistic inference of LLMs.

---

> > ### Author Rebuttal · Reviewer_eN7h · 2026-04-04
> >
> > Thank you for clarifying my questions and addressing my concerns. I will raise my score.

---

> > > ### Author Response · Authors · 2026-04-05
> > >
> > > We would like to express our sincere gratitude to the reviewer for the thoughtful feedback and for increasing the score. We are pleased that our rebuttal successfully addressed the initial concerns. We will incorporate the points discussed during the rebuttal process into our revised manuscript. Thank you once again for the valuable feedback that has helped enhance our work.

---

### Official Review · Reviewer_H4vw · 2026-03-14

**Soundness:** 3
**Presentation:** 3
**Significance:** 2
**Originality:** 2
**Overall Recommendation:** 4
**Confidence:** 4

**Summary:**

The paper introduces DRPBench, a benchmark for evaluating LLMs on static data race prediction in concurrent C programs. It gathers 1 003 programs derived from the SV‑COMP suite, including 226 racing programs with 549 manually annotated races and 777 race‑free programs. Results show a large performance gap: the best model achieves F1 about 74%, while most models perform poorly, revealing failure modes such as distraction by shared variables and misunderstandings of synchronization logic. The paper provides diagnostic analyses and suggests combining symbolic analyses with LLMs to improve precision.

**Compliance With Llm Reviewing Policy:**

Affirmed.

**Key Questions For Authors:**

Q1: In Table 1, the symbolic program analyzer Deagle appears to underperform compared with a majority of LLMs. However, the paper states that Deagle is a state-of-the-art static symbolic analyzer that has secured the championship in the SV-COMP competition for four consecutive years (2022–2025). The authors should therefore verify whether Deagle was used correctly and whether its configurations are appropriate.

**Limitations:**

Yes

**Strengths And Weaknesses:**

+ Benchmarking LLMs on concurrency analysis addresses an important research challenge and may guide future tool development.
+ The dataset and evaluation framework can be reused by other researchers to compare new models and prompting strategies.
+ The paper clearly motivates the problem, describes dataset construction, and explains evaluation metrics with concrete examples.


- The methodology, i.e., using static analysis and manual annotation, is similar to other LLM benchmarks; dataset novelty is modest since it is derived from existing SV‑COMP programs.
- The work does not show that LLMs outperform established static or dynamic race detection tools except Deagle, which shows rather low performance, so the benefits for practitioners are uncertain.

---

> ### Author Rebuttal · Authors · 2026-03-30
>
> We thank the reviewer for the constructive feedback and the opportunity to clarify the performance of Deagle. We address the concerns regarding its configuration and the performance gap as follows.
>
> ### **Verification of Configuration and Execution**
>
> We strictly followed the official "Usage in SV-COMP" documentation provided in the Deagle repository to ensure the tool was executed with the champion-grade settings used in the 2022–2025 competitions. We manually inspected the execution logs for every test case and confirmed that Deagle completed its analysis (or reached a timeout) without internal crashes or configuration errors, justifying the integrity of our evaluation.
>
> ### **Why Deagle Struggles on DRPBench**
>
> The performance gap between Deagle and LLMs is primarily due to the presence of **unconstrained loops**. As discussed in **Appendix D**, programs in DRPBench contain such loop structures that pose challenges for symbolic analyzers.
>
> Symbolic tools like Deagle rely on a heuristically pre-defined **unwind limit** to bound their analysis for loops, which denotes how many times they deem the loop should be unwound. If a data race is nested deep within complex loop logic, an insufficient limit may lead to false analysis results. Increasing this limit often leads to path explosions and timeouts rather than successful detection.
>
> Unlike symbolic tools that must formally prove reachability—often getting stuck on complex control flow—LLMs act as "heuristic pattern matchers." They can identify conflicting access patterns to shared variables even when the formal reachability of those paths is computationally expensive for symbolic detectors.
>
> By strictly following the "Usage in SV-COMP" documentation provided in the Deagle repository, we ensured that Deagle was evaluated under its optimal unwind limit configuration. The evaluation results on DRPBench reflect the inherent trade-offs between formal symbolic analysis and neural-based heuristics.
>
> We will integrate the above discussions into our manuscript in future revisions.

---

### Decision · Program_Chairs · 2026-04-30

**Decision:**

Accept (spotlight)

**Comment:**

This paper introduces DRPBench, a benchmark to evaluate the LLMs' ability to predict static data races in concurrent C programs. It requires the LLMs to identify the specific variable and exact lines that are affected by the potential race conditions. A wide range models are evaluated against this new benchmark, including Gemini-3, Claude-4.5, etc, and it identifies several common failure modes.

Overall, this papers presents an interesting benchmarks that tests the reasoning and coding capabilities of LLMs, and identifying race conditions are crucial in real-world software development as they are harder to identify by test suites. The limitations for this paper include limited language diversity (i.e., only C programs) and although there are previous work on detecting race conditions, this paper take a big step further to require the model to identify the exact places, which makes this task harder and less noisy. I'd recommend this paper to be accepted.